# A Review of Key Regulators of Steady-State and Ineffective Erythropoiesis

**DOI:** 10.3390/jcm13092585

**Published:** 2024-04-27

**Authors:** Ioana Țichil, Ileana Mitre, Mihnea Tudor Zdrenghea, Anca Simona Bojan, Ciprian Ionuț Tomuleasa, Diana Cenariu

**Affiliations:** 1Faculty of Medicine, University of Medicine and Pharmacy “Iuliu Hatieganu”, 8 Victor Babes Street, 400012 Cluj-Napoca, Romania; ilmitre@yahoo.com (I.M.); mzdrenghea@umfcluj.ro (M.T.Z.); simona.bojan@umfcluj.ro (A.S.B.); ciprian.tomuleasa@gmail.com (C.I.T.); diacenariu@gmail.com (D.C.); 2Department of Haematology, “Ion Chiricuta” Institute of Oncology, 34–36 Republicii Street, 400015 Cluj-Napoca, Romania; 3MEDFUTURE—Research Centre for Advanced Medicine, 8 Louis Pasteur Street, 400347 Cluj-Napoca, Romania

**Keywords:** erythropoietin, gene expression, iron metabolism, anemia, inflammation, stress erythropoiesis, gene therapy, microRNAs

## Abstract

Erythropoiesis is initiated with the transformation of multipotent hematopoietic stem cells into committed erythroid progenitor cells in the erythroblastic islands of the bone marrow in adults. These cells undergo several stages of differentiation, including erythroblast formation, normoblast formation, and finally, the expulsion of the nucleus to form mature red blood cells. The erythropoietin (EPO) pathway, which is activated by hypoxia, induces stimulation of the erythroid progenitor cells and the promotion of their proliferation and survival as well as maturation and hemoglobin synthesis. The regulation of erythropoiesis is a complex and dynamic interaction of a myriad of factors, such as transcription factors (GATA-1, STAT5), cytokines (IL-3, IL-6, IL-11), iron metabolism and cell cycle regulators. Multiple microRNAs are involved in erythropoiesis, mediating cell growth and development, regulating oxidative stress, erythrocyte maturation and differentiation, hemoglobin synthesis, transferrin function and iron homeostasis. This review aims to explore the physiology of steady-state erythropoiesis and to outline key mechanisms involved in ineffective erythropoiesis linked to anemia, chronic inflammation, stress, and hematological malignancies. Studying aberrations in erythropoiesis in various diseases allows a more in-depth understanding of the heterogeneity within erythroid populations and the development of gene therapies to treat hematological disorders.

## 1. Introduction

Hematopoiesis is the formation of mature blood cells; red blood cells, white blood cells and platelets from a common progenitor called a hematopoietic stem cell (HSC). Hematopoiesis is an active and continuous process throughout a person’s lifetime. The blood cells differentiation hierarchy starts from HSC, each cell type being derived from their own progenitor cell (red blood cells from committed erythroid progenitors, T-cells, B-cells, and natural killer cells from lymphoid progenitors and granulocytes, megakaryocytes, monocytes, and macrophages from common myeloid progenitors). The process of hematopoiesis is characterized by five branches of hematopoiesis: erythropoiesis, lymphopoiesis, granulopoiesis, monopoiesis, and thrombopoiesis. Initially, two major progenitor pathways originate from HSC: the common myeloid progenitor (CMP) and the common lymphoid progenitor (CLP). Committed erythroid progenitors support the production of 2 million erythrocytes per second in human adults via a synchronized regulation of iron-heme biosynthesis through hormones hepcidin and erythroferrone, amino acid-induced mTOR signaling and glucose metabolism, indispensable for de novo nucleotide biosynthesis [1,2].

## 2. Steady-State Erythropoiesis

Normal erythropoiesis takes place in the bone marrow of healthy adults by full maturation of erythrocytes from proerythroblasts. During this process, the cells undergo many stages of differentiation; the CMP subsequently divides into a granulocyte-monocyte progenitor (GMP) or a megakaryocyte-erythroid progenitor (MEP). The burst-forming unit erythroid (BFU-E) and colony-forming unit erythroid (CFU-E) cells are the committed erythroid progenitors traditionally defined by their ability to form colonies. The proerythroblast (ProE) is the earliest morphologically recognizable stage of an erythroid precursor and can be isolated by surface staining of the transferrin receptor (CD71). Further on, the ProE undergoes sequential differentiation into the basophilic, polychromatophilic, and orthochromatic erythroblast stages, ultimately leading to enucleation and the formation of a reticulocyte [3,4]. Upon entering the bloodstream, the reticulocyte undergoes maturation through the remodeling of its plasma membrane, the loss of internal organelles, and ultimately transforms into a red blood cell (Figure 1).

### 2.1. Erythroblastic Islands

The natural progression and maturation of red blood cells within the bone marrow are controlled by the supportive microenvironment, comprised of macrophages and erythroblasts, within structures known as erythroblastic islands (EBI). These functional units can be identified through conventional and specific immunological stains in bone marrow biopsies. First observed by Marcel Bessis in 1958, the erythroblastic island serves as the primary location for terminal erythropoiesis in mammals [5]. While these islands are primarily located in the bone marrow during regular erythropoiesis, they extend to the fetal liver and adult spleen during stress erythropoiesis. This expansion occurs when there is a rapid generation of erythrocytes in response to inflammation and anemia [5,6].

The central macrophage within the erythroblastic island plays a crucial role in a highly specialized process, which involves the ingestion of the enucleated nucleus and the subsequent recycling of nucleotides following nuclei degradation [7]. Macrophages have been found to promote proliferation and survival of erythroblasts in in vitro [8] and in vivo [9,10] conditions, particularly in a stressful environment, i.e., during immune challenge.

Within the erythroblastic islands of the bone marrow, late-stage erythroid precursor cells express FAS ligand. This ligand has the potential to interact with early erythroid precursors leading to caspase activation, subsequently inducing apoptosis and halting maturation. When there is a significant demand to generate new erythroid cells due to blood loss from bleeding or hemolysis, erythropoietin opposes this process, enabling the cells to survive and mature, even in the presence of numerous late erythroid precursors [7]. During erythropoiesis, activated caspases cleave their primary natural targets within the nucleus, including GATA-1, which amplifies cell death and impedes the process of erythroid differentiation [11,12,13].

Studying the mechanisms in the erythroblastic island niche during normal and stress-induced erythropoiesis has been performed through in vivo experiments in mice, with complexities further analyzed using straightforward in vitro systems. Advancements in stem cell technologies and gene editing have allowed for a more comprehensive examination of the human niche [5].

### 2.2. Regulation of Steady-State Erythropoiesis

Kidneys respond to low levels of oxygen by releasing the hormone erythropoietin (EPO), which triggers erythropoiesis [14]. During maturation, the proerythroblast becomes smaller in size, organelles are lost, and the nucleus is expelled from the cell. Immature reticulocytes are discharged into the circulation from the bone marrow. The quantity of reticulocytes present in the peripheral blood serves as an indicator of the erythropoiesis rate in the bone marrow.

The generation of red blood cells is an intricate system that relies on oxygen sensors, cytokines (EPO), and various factors, including regulators of iron metabolism. These components collectively regulate both steady-state and stress-induced erythropoiesis, ensuring the adequate supply of oxygen to peripheral tissues and stable hemoglobin concentrations [11].

An intricate system of physiological communication pathways and networks is accountable for the generation, distribution, and replacement of red blood cells in people with good health [11,15,16,17]. These networks maintain hemoglobin concentrations at a consistently stable level throughout a person’s life. The process of erythropoiesis initiates in the bone marrow through the commitment of pluripotent myeloid progenitor cells. The subsequent differentiation into immature erythroid progenitors preserves the specific proliferative capacity. In turn, the progenitor cells go through additional differentiation and maturity. The action is controlled by a complex system of transcription factors and epigenetic regulators [18,19]

Erythropoietin serves as the primary cytokine regulator for red cell production, operating at the late erythroid progenitor level through a homodimeric receptor. This receptor initiates JAK2 kinase activity, leading to subsequent STAT5 activation [20,21,22]. In vivo and in vitro models show that erythropoietin leads to STAT5 phosphorylation in early erythroblasts, which is a key mechanism in steady-state erythropoiesis and in stress conditions such as hypoxia [23]. Erythroid precursors in the bone marrow with lower sensitivity undergo apoptosis upon caspase activation when erythropoietin levels are low, whereas at higher erythropoietin concentrations, most cells manage to survive and undergo differentiation [24,25].

In healthy individuals, early-stage erythropoiesis is represented by the erythroid differentiation pathway from MEP to Eps (erythroid progenitors), and late–stage erythropoiesis as the final maturation stage into enucleated RBCs, which do not proliferate anymore [26]. Late-stage definitive erythroid cells rely on both erythropoietin (EPO) and its receptor (EPO-R) for survival, proliferation, and terminal maturation of primitive erythroid precursors.

GATA-1 has a crucial role in determining the lineage commitment, differentiation, and survival of erythroid progenitors. Specifically, GATA-1 initiates erythropoiesis by controlling the transcription of various genes related to erythroid differentiation. These include genes associated with heme and globin synthesis, glycophorins, BH-3 anti-apoptotic genes, genes regulating the cell cycle, and the gene encoding the erythropoietin receptor (EPOR) [27,28,29]. Erythropoietin primarily affects myeloid precursor cells to promote their survival, facilitating the erythroid differentiation process mostly driven by GATA-1.

Key molecular participants in these networks encompass iron, regulators of iron metabolism like transferrin receptors-1 and -2, as well as early acting hematopoietic growth factors including stem cell factor and interleukin-3. Additionally, classical hormones like thyroid hormones, androgens, corticosteroids, activin/inhibin, and others are integral components [30,31,32,33,34].

Numerous kinase-signaling cascades have been thoroughly studied, encompassing pathways such as Janus kinase/signal transducer and activator of transcription (JAK/STATs), PI3K/Akt, and mitogen-activated protein kinases (MAPKs). These signaling pathways are involved in the differentiation of hematopoietic stem and progenitor cells (HSPCs) into mature red blood cells, by phosphorylating multiple substrates, such as extracellular signal related kinases (ERK), p38 kinase-dependent pathways and phosphoinositide 3-kinase (PI3K) [35]. Coulon et al. enhanced cellular responsiveness to EPO through pIgA1 and TfR1 (also known as CD71), two regulators involved in the activation of MAPK and PI3K signaling pathways, with the well-defined role of stimulating erythropoiesis [36].

With aging, there is a decline in the quantity of erythroid islands within the bone marrow, accompanied by an enlargement in their size. The reduction in the numbers of stem cells and erythroid progenitors during the aging process might be counteracted by an enhanced proliferation of local erythroid progenitors [1,37,38].

The generation of new erythrocytes within the bone marrow niche is a dynamic process that involves the stimulation of several components, including vascular endothelial cells, osteoblasts, hematopoietic cells, stromal cells, and the extracellular matrix [39]. There is a constant interaction in the bone marrow microenvironment between the hematopoietic cells and cell adhesion molecules, growth factors, and cytokines including insulin-like growth factor 1 (IGF-1), interleukin-3 (IL-3), granulocyte-macrophage CSF (GM-CSF), along with EPO and stem cell factor, in order for the erythroid cells to develop and differentiate.

Different cellular and molecular pathways associated with the production and maturation of red blood cells have been identified and characterized: Stem Cell Factor (SCF), c-KIT receptors, MAPK family members, IL-3 stimulates PI3K, RAS/MAPK, interleukin-6 (IL-6)—IL-6, IL-11, LIF, and OSM, erythropoietin (EPO) [35]. We can only leverage our comprehension of kinase signaling in normal erythropoiesis to achieve clinical benefits in various types of anemias once we have a more profound understanding of the process.

These in vivo models [40,41,42] could be employed to simulate red blood cell diseases and identify novel drug targets, offering significant advantages for patients resistant to existing treatments. Moreover, these models might contribute to research on the in vitro production of red blood cells, offering a potential alternative to the prevalent practice of donor blood transfusions in treating blood disorders.

### 2.3. MicroRNAs Involved in Steady-State Erythropoiesis

MicroRNAs are small non-coding, single-stranded RNAs (19–24 nucleotides long), that negatively regulate gene expression through mRNA degradation at the translational level [43]. MicroRNAs are generally considered key regulators of cell proliferation, differentiation, development and apoptosis [44]. There are several microRNAs involved in steady state erythropoiesis mediating the growth and development of normal erythroid cells (miR-126), regulating oxidative stress, erythrocyte maturation and differentiation (miR-210, miR-362, miR-188), hemoglobin synthesis (miR-144/451, miR-486-3p, miR326), transferrin function (miR-320) and iron homeostasis (miR-122). miR-144 and miR-451 are involved in maintaining erythroid homeostasis while the down-regulation of miR-221, miR-222 and miR-223 is required for terminal differentiation and proliferation [45,46].

#### 2.3.1. Erythroid Differentiation

In vivo and in vitro models show that increased expression of miR-486-5p stimulates differentiation and survival of normal CD34(+) erythroid cells by targeting FOXO1 and PTEN genes, significant up-regulation has been observed in chronic myeloid leukemia (CML) [47]. MiR-23a and miR-27a induce GATA1 depended erythroid differentiation in human hematopoietic CD34(+) progenitors as well as mice and zebrafish experimental models [48,49]. Andolfo et al. show in an experiment on CD34(+), K562 and HEL cells that miR-Let-7d targets the DMT1-IRE gene and impairs erythroid differentiation through the disruption of the iron metabolism [50]. MiR-181 promotes erythroid differentiation in experimental models of CD34(+) cells by repressing Lin28 expression and disrupting the Lin28-let-7 [51]. Down-regulation of miR-150, miR-155, miR-221, miR-222 has been demonstrated as necessary for both early and late erythroid proliferation, while up-regulation of miR-451, miR-24, and miR-16 has been observed [44]. Wang et al. show that overexpression of miR-376a detected in K562 cells silences CDK2 and Ago2 and inhibits erythroid differentiation [52].

#### 2.3.2. Erythroid Maturation

MiR-144/451 is up-regulated after erythroid differentiation, it targets the FOXO3, GATA2 and KLFD genes and in addition to erythroid maturation it regulates oxidative stress and hemoglobin synthesis [44,53,54,55]. During the maturation of erythroid cells, overexpression of miR-191 blocks enucleation by targeting Riok3 and Mxi1 genes [56]. Choong et al. in an experiment on CD34(+) and K562 cells derived from human umbilical cord blood concluded that miR-22, miR-28, miR-185 correlate with CD71, CD36, and CD235a expression and act in erythroid maturation by targeting TLK, H3F3B, MAP3K3, BCL9L and TYRO3 genes [57]. Additionally, their work associated the miR-181 family, miR-221 and miR-154 with common myeloid/erythroid progenitor commitment and miR-32, miR-136, miR-137 with early erythroid commitment [57]. Rivkin et al. demonstrated in an in vivo experiment using a mouse loss-of-function allele model that miR-142 plays a critical role in erythrocyte maturity along with the regulation of erythrocyte size, function and survival. Furthermore, its control on the ACTIV network helps regulate membrane skeleton organization [58].

## 3. Ineffective Erythropoiesis

In certain pathological states, this regulatory network becomes overburdened or dysfunctional, leading to either polycythemia or anemia. Over a billion and a half people around the globe are affected by anemia, bringing the total worldwide disease burden at around 9% of patients with poor numbers of healthy RBCs and low blood hemoglobin [59]. Disorders affecting red blood cells encompass a wide range of conditions, from hereditary disorders like thalassemia and sickle cell anemia to acquired disorders such as polycythemia vera and paroxysmal nocturnal hemoglobinuria (PNH), which can result in clinical outcomes varying from mild to fatal.

### 3.1. Anemia

The World Health Organization (WHO) established specific criteria, nearly a half century ago, to define anemia based on hemoglobin (Hb) levels in the blood, less than 12.0 g per deciliter (g/dL) for women and less than 13.0 g/dL for men [60]. In recent years, anemia is categorized into subtypes based on mean cellular volume (MCV) and by the reticulocyte index (RI), accounting for the bone marrow’s responsiveness to anemia. A low RI suggests inadequate compensation, while a high RI indicates active efforts to address the underlying cause of anemia, whether by increased destruction of red blood cells or recovery from a previous episode of anemia [61]. There are four major types of anemia, classified by MCV, as presented in Table 1 [62].

Often, anemia manifests as a complication in patients battling cancer and more specifically hematologic malignancies originating from diverse factors like neoplastic cell infiltration in the bone marrow, hemolysis, poor nourishment, and ineffective erythropoiesis caused by diseases like chronic lymphocytic leukemia, multiple myeloma, myelodysplastic syndromes, thalassemia, chronic kidney disease, hemorrhage or cytotoxic therapies, chemotherapy agents, CAR T-cell and other immune-based therapies. Anemia can be regarded as a prognostic tool in some hematological malignancies or a result of the disease outcome [63] in others. One such example is clonal hematopoiesis of indeterminate potential (CHIP), a hematological condition marked by the expansion of hematopoietic clones triggered by somatic mutations in stem and progenitor cells, common in older adults, without evident bone marrow disorders [64].

#### 3.1.1. Thalassemias

Thalassemia syndromes are severe health-threatening conditions. The mutations can be expressed in the α (HBA1/HBA2) and β globin (HBB) genes, inherited in an autosomal recessive manner [65]. α thalassemia is an inherited disorder characterized by reduced or absent synthesis of alpha-globin chains caused by somatic mutations of the ATRX gene that may result in neoplastic transformation of the bone-marrow progenitors [66]. β-Thalassemia is a genetic disorder with decreased synthesis of beta-globin chains, caused by point mutations in the HBB gene, which are essential components of hemoglobin. This mutation along with the arrest of maturation mediated by the TGF β superfamily leads to ineffective erythropoiesis, a high rate of proliferation and apoptosis, extramedullary hematopoiesis, and severe anemia [67].

#### 3.1.2. Congenital Dyserythropoietic Anemia (CDA)

Another group of rare inherited disorders produced by an ineffective erythropoiesis are congenital dyserythropoietic anemias (CDAs). CDAs are caused by mutations in genes involved in the process of erythropoiesis. For example, CDA type I is associated with mutations in the CDAN1 gene, CDA type II with mutations in the SEC23B gene, CDA type III with mutations in the KLF1 gene and CDA type IV is caused by a unique heterozygous variant c.973G>A (p.E325K) in the KLF1 gene. The main symptoms of CDAs are hemolytic anemia, iron overload, aplastic crisis, splenomegaly, cirrhosis, gallstones, and skeletal abnormalities [68].

#### 3.1.3. Anemia of Inflammation and Chronic Disease

Anemia linked to chronic inflammatory diseases induces tissue hypoxia, irrespective of its source. This condition triggers increased erythropoietin (EPO) production by the kidneys, subsequently stimulating erythropoiesis in the bone marrow. Conversely, under instances of intense anemic stress, an alternative stress erythropoiesis pathway is activated to ensure the delivery of oxygen to the tissues. Inherited forms of anemia, like thalassemia, and acquired types like MDS—due to inefficient production of red blood cell precursors—are connected to abnormalities in the later stages of erythropoiesis [13].

Chronic inflammatory conditions, prolonged infections, autoimmune diseases, and cancer lead to an increased production of pro-inflammatory cytokines that will negatively influence physiological erythropoiesis, and on top of that, they will generate the synthesis of myeloid cells. In these conditions, acute anemia occurs, and the stress erythropoiesis pathway is activated through the appearance and mobilization of stress erythroid progenitors (SEPs).

SEPs originate directly from short-term reconstituting hematopoietic stem cells (ST-HSCs), in the adult spleen and liver of mice and were identified by CD34+Kit+Sca1+Lin− murine markers. These HSCs migrate from the bone marrow to the spleen, where they are identified by hedgehog (HH) and bone morphogenetic protein 4 (BMP4) signaling as SEPs. Three distinct populations of prematurely SEPs are identified in vitro as stress erythroid progenitors by the cell-surface markers that indicate increasing maturity: CD34+CD133+Kit+Sca1+, CD34−CD133+Kit+Sca1+, and CD34−CD133−Kit+Sca1+ [69]. Each of these groups is capable of transplantation and self-renewal. However, despite the presence of stem cell markers, they are constrained to erythroid differentiation [70]. Xiang and team were able to validate that cells within the human BM, as the ones in C57BL murine models used, have the capacity to produce BMP4-dependent stress erythroid burst-forming units when subjected in culture to conditions that mimic stress erythropoiesis. Erythroid progenitors constitute a diverse group of cells. In the future, it will be essential to comprehend the regulatory mechanisms that contribute to this diversity. Erythroid progenitors play a distinctive role as a vital bridge between hematopoietic stem cells and terminal erythroblasts which are predominantly pre-programmed. As cells undergo terminal erythroid differentiation, they exhibit minimal to no capacity for proliferation and self-renewal [71].

Inflammatory signals play a significant role in enhancing NF-κB activity and promote the synthesis of pro-inflammatory cytokines, interferons type I (IFNα and IFNβ), type II (IFNγ) and tumor necrosis factor α (TNFα) with a direct impact on early differentiation and maturation of HSCs [72]. A crucial role in boosting self-renewal and differentiation properties of HSCs is attributed to mesenchymal stem cells (MSCs) by IL-6 and IL-1 cytokines release as a response to infection and inflammation [73,74]. Human granulocyte colony stimulating factor (G-CSF), is a protein consisting of 175 amino acids, produced by diverse cell types, including monocytes, fibroblasts, macrophages, and stromal cells [75]. It is an important player in the regulation of hematopoiesis and immunity, as it stimulates the BM to produce neutrophils and stem cells and release them into the bloodstream [76,77]. HSPCs mobilization in response to G-CSF was demonstrated in mice with homozygous deletion of the C3 gene [78,79].

Other inflammatory cytokines, including IL-6, IL-3, IL-12, and GM-CSF join G-CSF in its mobilizing activity of hematopoietic stem/progenitor cells (HSPCs) into the blood. However, the expansion HSCs is not linked to increased HSC activity. Mice treated with G-CSF exhibit notably lower repopulating activity in the bone marrow compared to untreated mice [80,81].

IL-1 (together with IL-1α and IL-1β genes) is the first interleukin identified from the IL-1 family (11 cytokines) engaged in the host immediate response to infections or inflammation [82]. IL-1 manifests many pleiotropic functions throughout the body as a lymphocyte-activating factor, in hemopoietin-1, osteoclast activation and secretion of metalloproteases, in fever development, and in maintaining homeostasis of the neuro-immuno-endocrine system. To prove this concept, Horai et al. [83] obtained and operated on KO mice carrying a null mutation in one of the IL-1α, IL-1β, or IL-1ra genes. The scientists concluded that all these forms of Il-1 work together in a regulatory milieu that controls fever development and glucocorticoid synthesis in normal physiology and under stress conditions. During acute need, interleukin 1 (IL-1) triggers rapid myeloid recovery in vitro and in vivo and increases HSC differentiation towards MMP by activation of PU.1 transcription factor, together with its target genes GM-CSF and M−CSF. In vitro IL-1β-induced stimulation of HSCs produced an increase in c-Kit + progenitors and mature myeloid cells [84]. In chronic exposure to injury and inflammation, HSCs lose part of their self-renewal capacity, with fewer colonies formed as a result of reduced clonogenic capacity of the progenitor cells in the presence of IL-1 due to a higher degree of HSCs maturation [84]. It can be concluded that IL-1 plays a double-edged sword influence in the BM microenvironment, in the process of myeloid differentiation of HSPCs.

Tumor necrosis factor-α (TNF-α) is an important member of the TNF family and a potent pro-inflammatory cytokine. Tumor necrosis factor alpha (TNFα) was initially obtained from serum and being predominantly expressed by primitive neutrophils and to a smaller extent by lymphocytes, NK cells, and endothelial cells [85]. HSCs maintenance, growth and differentiation in the BM niche is inhibited by tumor necrosis factor-α (TNF-α) and interferon-γ (IFN-γ) as well as their influence in shortening the life cycle of mature circulating erythrocytes [70].

PU.1 is an important regulator of myeloid differentiation in normal HSPC homeostasis. Transcription factor PU.1 (purine-rich DNA binding, SPI1) can be induced by TNF, via improved mRNA transcription and translation. This transcription is directly adjusted by NF-κB signaling and post-transcriptional degradation via microRNA-155 (miR-155). Long-term expression of PU.1 transcription factor results in myeloid commitment [86].

### 3.2. Hematological Neoplasms

Hematological neoplasms can be arranged by WHO5 Classification 2017 [87] into myeloid, lymphoid and myeloid/lymphoid neoplasms. Myeloid neoplasms originate from progenitor cells within the bone marrow that have the capacity to differentiate into various mature blood cell types, including erythrocytes, granulocytes, monocytes, and megakaryocytes. These progenitor cells undergo genetic alterations that drive abnormal proliferation and differentiation, leading to the development of myeloid neoplasms, namely: Myeloproliferative neoplasms (MPNs), Chronic myeloid leukemia (CML), Acute myeloid leukemia (AML), and Myelodysplastic neoplasms/syndromes (MDS). Lymphoid neoplasms include acute lymphoblastic leukemia, mature B cell neoplasms, mature T cell neoplasms and Hodgkin lymphoma. Myeloid/lymphoid neoplasms encompass eosinophilia and tyrosine kinase gene fusions, mastocytosis, dendritic neoplasms, mixed myeloid and lymphoid neoplasms.

MPNs are a group of hematological disorders characterized by the abnormal production of myeloid cells within the peripheral blood during hematopoiesis in the bone marrow environment, genetic mutations, growth factors and transcription factor dysregulation [88]. The four major types of myeloproliferative neoplasms according to the WHO classification are Chronic Myeloid Leukemia (CML), Polycythemia Vera (PV), Essential Thrombocythemia (ET), and Primary Myelofibrosis (PMF).

#### 3.2.1. Chronic Myeloid Leukemia (CML)

CML is a myeloproliferative neoplasm mostly composed of proliferating granulocytes and Philadelphia chromosome t(9;22)(q34;q11) translocation (PMID: 32239758) that manifests in 15% of newly diagnosed cases of leukemia in adults. Patients can be asymptomatic or present rare manifestations like bleeding, thrombosis, gouty arthritis, retinal hemorrhages, and upper gastrointestinal ulceration [59].

#### 3.2.2. Acute Myeloid Leukemia (AML)

AML is a malignancy of the bone marrow, a disorder of hematopoietic stem cells of the myeloid lineage, characterized by clonal overexpansion [89]. It represents one of the most commonly diagnosed types of leukemia in adults and accounts for 1% of all cancers [59].

#### 3.2.3. Acute Erythroid Leukemia (AEL)

AEL is a rare subtype (2–5%) of acute myeloid leukemia characterized by a predominant proliferation of erythroid precursors in the bone marrow, leading to bone marrow failure and a poor prognosis compared to other subtypes of AML. AEL is a representative case of dyserythropoiesis, as it can cause up to 80% of immature erythroid precursors in the bone marrow. In 2001, the WHO presented two categories of AEL: the first subtype M6a (50% or more of erythroid precursor and 30% or more of blasts), and the second M6b (80% are immature erythroid precursors). This classification raised much debate and subsequently the 2016 WHO report reassigned type M6a to MSD leaving type M6b the only remaining subtype of AML [90].

#### 3.2.4. Polycythemia Vera (PV)

PV is a chronic myeloproliferative blood disorder where there is an overproduction of red blood cells (erythrocytosis) along with increased numbers of white blood cells and platelets. This results in thickened blood and reduced blood flow through small vessels, leading to complications such as blood clotting, increased risk of stroke or heart attack, and splenomegaly. The JAK2 mutation plays a central role in the pathogenesis of PV, driving the uncontrolled proliferation of blood cells [91].

#### 3.2.5. Myelodysplastic Syndromes (MDS)

MDS are clonal bone-marrow diseases of the elderly characterized by chronic cytopenias and morphologic dysplasia of hematopoietic cells with a high risk of progression to acute myeloid leukemia (AML) [92]. An ineffective erythropoiesis in MDS is a result of an inflammatory environment that induces malignant clonal alterations due to: chromosomal abnormalities induced by del(5q), del(7q) deletions/additions, specific mutations of the spliceosome (SF3B1, SRSF2), transcription factors (RUNX1, ETV6) [93], NLRP3 inflammasome activation, overexpressed SMAD2/3 downstream mediators, TGF-b signaling, epigenetic modifiers (TET2, DNMT3A 5, IDH1/2, ASXL1), RNA splicing factors, all drivers of MDS pathogenesis by promoting an inhibitory activity on RBCs maturation [92,94].

Ineffective erythropoiesis is one hallmark of MDS. ARC (Absolute reticulocyte count) in peripheral blood was found to be a biomarker (when ARC < 20 × 10^9^/L) together with a shorter overall survival in the evaluation of the severity of ineffective erythropoiesis in 776 MDS patients. Huang et al. concluded that the inadequate production of red blood cells in MDS may be partly due to premature ageing and apoptosis during erythroid differentiation as well as high-risk molecular genetic aberrations due to the altered expression of the ERCC1 gene [95].

### 3.3. MicroRNAs Involved in Ineffective Erythropoiesis

Mir-155 up-regulates the production of inflammatory cytokines G-CSF and TNFα by bone marrow stromal cells through the activation of NF-kB [96]. L. Wang et al. concluded that the Notch/miR-155/κB-Ras1/NF-κB pathway controls the inflammatory condition of the bone marrow environment and influences the progression of myeloproliferative diseases. Bašová P. et al. succeeded to induce increased levels of PU.1 in an AML (Acute myeloid leukemia) mouse model by administrating a combination of three therapeutic agents: AZA (5-Azacytidine), CEL (Celastrol), and AM155 (anti-miR-155), which inhibited the growth of myeloid malignant cells and prolonged the survival of mice with AML [86]. PU.1 directly controls the expression of miR-155 and other miRs as well, namely miR-146a, miR-342 and miR-338. In a CLL (Chronic lymphocytic leukemia) Eμ-TCL1 transgenic mouse model, the suppression of miR-155 (or miR-26A or miR-130a) leads to the induction of apoptosis [97]. TNFα, on the other hand, triggers apoptosis by the overexpression of Fas in bone marrow CD34 + cells. The administration of an anti-TNFα monoclonal antibody lowers anemia levels in human TNFα transgenic mice, by reducing the apoptotic erythroblasts [98].

The following miRNAs inhibit erythroid differentiation in K562 cells (miR-124, miR-200, miR-223, miR-224), in TF-1 cells (miR-200A, miR-221/222), in polycythemia vera (miR-16-2) and in other diseases, see Table 2.

## 4. Therapeutic Approaches

There are some approved drugs that stimulate erythropoiesis; one such drug is Epoetin Alfa (human erythropoietin), which is routinely used in the treatment of anemias associated with chronic kidney disease and cancer. Following these treatments, only 20% of patients reach normal hemoglobin values while 40% remain anemic [5]. The main signaling mechanisms essential for erythroid progenitor survival, proliferation and differentiation attributed to EPO/EPO-R interaction have been demonstrated to be JAK2/STAT5 pathway activation and phosphorylation [20,123]. Currently, several clinical trials are ongoing which aim to explore the treatment of anemia in patients with chronic kidney disease and cancer patients receiving chemotherapy using Peginesatide, a functional analog of erythropoietin [59].

In the case of thalassemia, most common treatments include blood transfusions, allogeneic stem cell transplantation, luspatercept, 5-azacytidine, decytabine and butyrate derivatives. In 2024, the FDA approved exagamglogene autotemcel (exa-cel), depending on the severity [67]. CDA treatment options include blood transfusions, hematopoietic stem cell transplantation (HSCT), interferon-α and drugs (luspatercept, sotatercept) that target ineffective erythropoiesis, new approaches on drugs that target both the anemia and the iron overload [68].

Several therapeutic options are being explored to regulate miRNA biogenesis such as CRISPR/Cas9-base genome editing, antagomirs, miRNA sponges and Small Molecules Inhibitors of miRNAs (SMIRs). Oligonucleotide analogs may be used to correct miRNA loss-of-function and anti-miRNAs oligonucleotides (AMOs or antagomiRs) as well as artificial miRNAs which are designed to silence specific target genes. These approaches have successfully been integrated into cancer research and hold promise for the development of erythropoietic stimulating agents [124,125,126].

Gene therapies that aim to stimulate erythropoiesis and to correct genetic defects that cause inherited forms of anemia, such as β-thalassemia and sickle cell disease, aim at repairing or replacing defective genes in hematopoietic stem cells (HSCs). CRISPR/Cas editing of the ß-globin gene has been explored. Histone deacetylases inhibitors like vorinostat and hypomethylating agents help regulate gene expression through chromatin modulation. DNA-methyl transferases such as DNMT1, BCL11A and decitabine stabilize DNA methylation marks during cell division. Small molecule drugs targeting pathways involved in erythropoiesis have been explored for the treatment of anemia. Erythroferrone (ERFE) is involved in the regulation of hepcidin expression and could potentially be considered as a target marker in ineffective erythropoiesis. SLN124, conjugated 19-mer short interfering RNA that targets the TMPRSS6 gene that has been recently associated with increased hepcidin levels in healthy adults [127]. Currently, JAK inhibitors like ruxolitinib have FDA approval for use in myelofibrosis [128,129,130].

Certain red blood cell disorders, such as paroxysmal nocturnal hemoglobinuria (PNH) and acquired aplastic anemia, can be successfully treated through bone marrow transplantation [131,132]. However, this intervention is linked to notable morbidity and mortality rates due to the necessity for preconditioning, a process that eliminates the host’s own hematopoiesis.

Efficient therapeutic options in MDS are few: erythropoietin and hypomethylating agents (Azacitidine and Decitabine), immunomodulating agents (Lenalidomide) and recently, luspatercept (a TGFb-pathway activin receptor trap) [93]. In the clinical trials (PACE-MDS phase 2) and (MEDALIST phase 3) approved by FDA in 2020, luspatercept exhibited encouraging results in the treatment of anemia in patients with transfusion-dependent lower-risk myelodysplastic syndrome (MDS) with ring sideroblasts, by promoting erythroid maturation [26].

## 5. Conclusions

The study of the complex interaction of signaling pathways, genetic, and epigenetic factors that regulate erythropoiesis is essential for understanding how erythropoiesis becomes disrupted in different hematological disorders and provides novel approaches for treatment. Ongoing research is revealing new aspects of these factors and their functions in maintaining erythroid homeostasis.

## Figures and Tables

**Figure 1 jcm-13-02585-f001:**
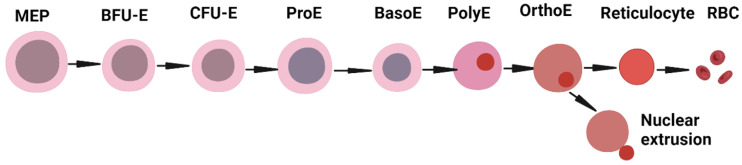
Steady-state erythropoiesis. Differentiation of blood cells from hematopoietic progenitor cells to the formation of mature red blood cells in peripheral blood. Created with BioRender.com (accessed on 17 April 2024).

**Table 1 jcm-13-02585-t001:** Classification of anemia by MCV.

Type of Anemia	MCV	Etiology
Hypoproliferative Microcytic Anemia	MCV < 80 fL	iron-deficiency anemia sideroblastic anemia thalassemia lead toxicity
Hypoproliferative Normocytic Anemia	MCV 80–100 fL	anemia of chronic disease hemolytic anemia aplastic anemia myelofibrosis leukemia cancer metastases
Hypoproliferative Macrocytic Anemia	MCV > 100 fL	megaloblastic anemia pernicious anemia nonmegaloblastic anemia MDS hereditary spherocytosis liver disease hypothyroidism folate and vitamin B12 deficiency
Hemolytic anemia		extravascular hemolysis intravascular hemolysis

**Table 2 jcm-13-02585-t002:** MicroRNAs and their role in ineffective erythropoiesis.

mRNA	Target	Models	References
miR-150	c-Myb	Stimulates MEP differentiation in megakaryocytes, inhibiting erythropoiesis	[99]
miR-124	c-MYB TAL1	Inhibits erythroid differentiation K562 cells	[100]
miR-17/92 cluster	MAPK signaling	HbF (fetal hemoglobin regulation) regulation	[101]
miR-221/222	c-Kit	Inhibits erythropoiesis TF-1 cells,	[102]
miR-223	LMO2	Inhibits erythroid differentiation, K562 cells	[103,104]
miR-24	ALK4	Inhibits terminal differentiation K562 cells	[100]
miR-155	PU-1 ETS-1 CEBP β SHIP 1	Inhibits erythropoiesis Mouse and K562 cells	[44,105]
miR-150	Riok-3 Mxi-1	↓miR-150 pathophysiology of PV ↑miR-150 Chronic lymphocytic leukemia	[106,107]
miR-143↓ miR-145↓	ERK MAPK	B-cell malignancies	[108]
miR-16-2	c-MYB	Abnormal expansion of erythroid cells in polycythemia vera	[109]
miR-146b	NF-kB	Suppresses NF-κB activation	[110]
miR-146a	γ globin	Inhibits γ globin expression β-thalassemia	[111]
miR15a	c-Myb	Inhibits BFU-E transition to CFU-E	[49,112]
miR-16-1	c-Myb	Inhibits BFU-E transition to CFU-E K562 cells	[49,111]
miR-96	γ globin	Inhibits γ globin expression	[111,113]
miR-9	FOXO3	Inhibits erythrocyte differentiation in mice	[114]
miR-125b-1	TP53 MCL1 BAK1	Transcriptional activation of miR-125b-1 led to lymphoid precursor transformation, myelodysplasia and other types of leukemias	[115]
miR-125b-2	DICER1, ST18	↑DS-AMKL, ↑DS-TL, ↑non-DS-AMKL Pathogenesis of megakaryoblastic leukemia	[46]
miR-145	TIRAP Fli-1	Loss of miR-145 contributes to the pathogenesis of 5q- syndrome	[46]
↑miRs-126/126	PTPN9	Inhibition of erythropoiesis	[116]
↑miR-669m	Akap7, Slc22a4 Xk genes	Inhibited erythroid differentiation in mice	[117]
miR-200A	PDCD4 and THRB	↑miR200A in K562 and TF-1 cells, inhibited erythroid differentiation	[118]
↑miR-2355-5p	KLF6	↑miR-2355-5p in HUDEP-2 and CD34+ cells increases γ-globin synthesis by suppressing expression of KLF6 (important transcription factor in erythropoiesis)	[119]
miR-101-3p	Rac1, SUB1, TET2, and TRIM44	High proliferation in β-thalassemia/HbE erythroblasts	[120]
miR-24	ALK4 (activin type I receptor)	Inhibition of erythropoiesis	[101]
miR-320a	SMAR1, TFRC	Inhibits erythrocyte differentiation and apoptosis K562 cells	[121]
miR-451	GATA1, GATA 2, RAB14	K562 cells, mice erythroid cells Inhibits mitochondrial respiration	[122]
miR15a	c-Myb	Inhibits BFU-E transition to CFU-E	[49,112]

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
