# Peer review of "A Review of Key Regulators of Steady-State and Ineffective Erythropoiesis"

_jcm, 2024, doi:10.3390/jcm13092585_

Round 1

Reviewer 1 Report

Comments and Suggestions for Authors

The title and abstract entice the reader to read the review, which unfortunately does not live up to the announced expectations. The subject is interesting and the potential is high but the structure is difficult to decipher and the terminology is often sketchy and inexact.

Please find below my principal remarks:

“each cell type being derived from their own progenitor cell (red blood cells from erythrocytes, T-cells, B-cells, and natural killer cells from lymphocytes…” There is a big mistake or a very strange sentence structure: erythrocytes are not of course progenitor cells as lymphocytes, RBC and erythrocytes are synonymous, pay attention.

“five primary pathways of differentiation: erythropoiesis, lymphopoiesis, granulopoiesis, monopoiesis, and thrombopoiesis.” Pathways of differentiation is a very strange and unusual terminology, branches of hematopoiesis are more appropriate.

Please pay really attention to the terminology, blasts are at present used almost exclusively in pathological context, moreover there are several well-known differentiation steps between CMP and ProE.

“Is the hematopoiesis subunit process of red blood cells production, in the bone marrow of healthy adults, by full maturation of erythrocytes from proerythroblasts” Is that a question? Typing mistake? Anyway, it is an incomprehensible sentence, please correct and explain more clearly.

“from the BFU-E (Burst  forming units-erythroid) to the reticulocyte state take place within the erythroblastic island (EBI)”. Please consider that the scientific community is rather agreed to consider EBI as a macrophage surrounded by erythroblasts, CFU localization is debated but no papers show BFU in EBI. Moreover, recent papers revised the BFU/CFU classification desiccating more finely the heterogeneity of this compartment and talking of EP1/EP2/EP3/EP4.

Attention in the figure 1 lacks PolyE!!!!

“Reticulocytes are slightly basophilic” ? Can you explain the idea behind this sentence.

“early-stage erythropoiesis is characterized by proliferation of HSCs and their differentiation into erythroid progenitors” Pay attention at the terminology. When we talk of erythropoiesis we don’t talk of HSC proliferation. early erythropoiesis is the specific engagement in erythroid differentiation pathway from MEP to EPs.

Really pay attention to terminology and clarity of sentences, in the introduction you talk of progenitors and precursors often in inappropriate manner and without give a precise definition for the reader who could get lost. The same applies to EBI, in the same paragraph you talk of EBI, erythroblastic island niche, erythroid islands, erythroid blood islands…please homogenize terminology to help the reader.

“During erythropoiesis, activated caspases not only cleave their primary natural targets within the nucleus but also target GATA-1. This dual action amplifies cell death and impedes the process of erythroid differentiation” Please revise this sentence, the first part is in opposition with the second one or at least it gives this impression.

“Disfunctional (Inefficient) Erytropoiesis”: attention to the typing mistakes here!!!! And usual we talk more of diserythropoiesis or ineffective erythropoiesis and not of dysfunctional or inefficient.

“the total worldwide disease burden at around 9% of patients with poor numbers of healthy RBCs and low blood hemoglobin” Reference?

SEP are identified in mouse? CD34+Kit+Sca1+Lin− are murine markers. This part is difficult to read and understand. Before you talk of disease and then directly to SEP without an introduction.

“one such drug is erythropoietin (EPO)” but in the introduction you talk of EPO as cytokine, revise the sentence.

In view of the title in this subparagraph (Anemia), I expected a description of the types of anemias and how they impact the different stages of erythropoiesis and the signaling pathways. Moreover, there is not logical continuity: SEP than treatments and then age and BM, the reader is completely lost! Change the title or the content, anyway focus on one aspect that is really link to anemia. Please to inspire by the well-written and organized MDS subparagraph.

Strange approach for the 3.3, why put all together PV, MDS (that the authors discuss just before) and AEL? A paragraph on MPN (focus on PV and ET) and another on AEL are a more appropriate option.

That does it get to do here the “inflammation” when you talk on diseases? Very strange structure organization. Maybe put it in another paragraph talking about the biological signs of diserythropoiesis? Anyway, it is not coherent to keep here. Moreover, the inflammation part is turn on hematopoiesis and not at all on erythropoiesis, only the last sentence talks on anemia. The goal of the review is the erythropoiesis.

A small introduction and a clear definition of microRNA is suitable before getting to the heart of the subject. Very interesting part but it's a bit of a catalogue, it would be interesting to go a bit more in details.

Comments on the Quality of English Language

Often the sentences are not very clear or too laborious, please make an effort.

Author Response

Thank you for the review, please see attached our reply. 

Reviewer 2 Report

Comments and Suggestions for Authors

The review Is informative and  consise. The information are updated IT would have been morei informative  if the novel lines or treatment were more extensively Disscused.

Author Response

(The authors gave the same response as above.)

Reviewer 3 Report

Comments and Suggestions for Authors

Give more details about the mechanism involved in erythropoiesis in the abstract section.

Author Response

(The authors gave the same response as above.)

Reviewer 4 Report

Comments and Suggestions for Authors

General comments

Overall, the reviewer advises that the authors take the extensive work they've done to produce Tables 1 and 2, and reformat the review to expand on normal and ineffective erythropoiesis from the perspective of microRNA involvement. The authors clearly have extensive clinical knowledge, but have not articulated a convincing understanding of the genetic and epigenetic underpinnings of normal and ineffective erythropoiesis.

A review article should be a comprehensive report on the state of the literature and concepts contained within it for any given subject. Therefore, it is essential that the authors take care to ensure that the scope of the review (and its exclusions) are clearly stated, and that all other parameters are described comprehensively. 

Title needs revision. Signalling pathways are barely discussed. Instead, the review focuses on individual signalling molecules but does not describe how alterations in upstream or downstream pathway elements influence erythropoiesis. The only epigenetic regulatory molecules that are discussed in this review are microRNAs, that should be reflected in the title.

Flow is severely compromised by the extent to which the paragraphs seem to exist independently of one another – there is very little linking between concepts. Moreover, concepts are revisited in somewhat random orders throughout the sections.

Specific notes

·      Author contributions need revising to reflect that the paper is a review article and does not present novel experimental data nor analyses data from other studies.

·      First citation appears at line 52 after extensive descriptions of haematopoeisis but is not the most recent review on the process. Find most recent haematopoiesis review for citation, Oburoglu 2016 focuses on metabololic regulation of the process and is not recent.

·      Oburoglu 2016 is not the original paper in which the calculation of erythrocyte production per second. The most recent citation for this calculation is: J. Palis, "Primitive and definitive erythropoiesis in mammals," Frontiers in physiology (2014).

·      Line 54 – sentence is incomplete. Should begin with the words in the title and requires extensive editing of the English for clarity.

·      Line 55 – specify that the erythroblast island occurs in the bone marrow for clarity.

·      Line 58 – “The reticulocyte once entered in the bloodstream undergoes maturation by remodeling of its plasma membrane (loses internal organelles) and ultimately transforms into a red blood cell.” Information within parentheses does not explain the plasma membrane remodelling. 

·      Throughout – latin should be italicised (e.g., in vitro).

·      Line 65 – relevance of the mouse models cannot be ascertained without first:

o   Describing what clodronate liposomes do (trigger macrophage apoptosis when endocytosed).

o   Introduce the EMP-null C57BL/6J (erythroblast macrophage protein knockout on a black six background etc).

o   Describe CD169 macrophage cell surface receptor and why diptheria toxin results in cell death.

·      Line 66 – the authors reference abnormal macrophage differentiation, do they mean abnormal erythrocyte differentiation? If not, authors must link erythropoiesis defects to the perturbation of macrophages.

·      Line 68 – qualify statement about “particularly in a stressful environment”. I.e., during immune challenge? 

·      Line 70 – proerythroblasts are not properly described.

·      Line 70 - “(organelles without haemoglobin)” does not make sense.

·      Line 71 – description of organelle clearance and enucleation should be linked to the transition from proerythroblasts to later erythroid cell stages like basophilic, polychromatic and orthochromatic erythroblasts (as the authors show in Figure 1).

·      Line 79 – sentence beginning “RBCs maintain a balance” is incorrect. Perhaps the authors are referring to iron homeostasis where they say “stabilising hemoglobin concentrations”? Iron homeostasis, erythroid differentiation and haemoglobin biosynthesis are intimately linked, and defects in any of these processes results in protein and cellular imbalances which can be pathogenic. However, it is false to imply that RBCs maintain the balance.

·      An explanation of erythroblast islands, macrophages and their role in erythropoiesis is distributed through various parts of the review in a way that requires revision for flow and continuity. For example, the paragraph beginning at line 133 might be better placed immediately prior to line 64.

·      Subheadings could be applied to facilitate navigation through topics and make topic changes less jarring. E.g., a section on erythroblast islands, a section on microRNAs, etc.

·      Critical to mention the work of Merav Socolovsky’s group when discussing variable EPO sensitivity as it relates to STAT5. E.g., Porpiglia et al., 2012 (Plos).

·      Line 154 – the miRNAs mentioned should not come before introducing miRNAs as a concept (beginning line 157).

·      Line 162 – homeostasis should not be capitalised. Check that iron has been defined as Fe prior to use of term.

·      Line 165 – spelling mistakes. Dysfunctional. Erythropoiesis. Additionally, dysfunctional is not the same as inefficient. Especially considering that the authors then discuss polycythemia, which is over-production of erythrocytes and therefore directly contradicting the use of the term “inefficient”.

·      Line 169 – no citation provided for the statistic (9%). Imperative that proper citations are employed, particularly because the number of affected individuals varies depending on what metric is applied for thresholding.

·      Line 206 – paragraph beginning at line 206 is presented in written form but reads like a table. Incorrect capitalisations are used and no explanation or descriptions are provided. Consider revising into a table, providing a brief description, and updating the citation. Zivot is a fantastic and extensive review, but is inappropriately cited. E.g., Small molecule inhibitors of JAK already exist and are in trial/clinic, but the only mention of erythroferrone in Zivot is to say that it could be a small molecule target in the future. Moreover, ERFE is only relevant is a narrow range of disorders which should be defined. TMPRSS6 siRNAs are also relevant in this section.

·      Line 221 – paragraph beginning line 221 should be revised to fit into the title of the section and relate to the pre-ceding paragraph. I.e., preceding paragraph discusses a decline in erythroid islands with age, but the following paragraph discusses BM factors without linking them either to aging or anaemia.

·      Line 228 – title has been used as the start of the sentence. Sentence beginning line 229 does not make sense in its current form.

·      Line 252 – typo: survivor should be survival

·      Line 253 – typo: et all should be et al

·      Entire inflammation section needs significant revision. Authors should focus on how inflammation affects erythropoiesis, not haematopoiesis. For example, since all haematopoietic lineages are derived from a population of multipotential progenitor cells, when there is increased demand for one lineage, there must be reduced output of the others. Additionally, iron restriction in an environment of inflammation is referenced under section 3.1 (Anaemia), but would be worth including here.

·      Line 262 – typo: IL_6 should be IL-6

·      Line 280 – typo: et all should be et al

·      Line 306 – typo: et all should be et al

·      Section 3.3 – In the section discussing diseases of ineffective erythropoiesis, which appears in a paper that references genetics and epigenetics in the title, I would expect there to be an explanation of the genetic and epigenetic features underlying dyserythropoeisis for each of the diseases mentioned. Firstly, only MDS is discussed in any detail. Secondly, even in the case of MDS, the authors cite Huang et al., (2020), but fail to discuss the altered gene expression of ERCC1. Thirdly, thalassaemias and congenital dyserythropoietic anaemias would be relevant to this section - it is unclear why they haven't been addressed. 

·      Line 322 – typo: inhibits should be inhibit

Comments on the Quality of English Language

·      Extensive editing for clarity required with regard to use of English.

·      Moderate copy editing required.

·      Structure needs significant revision throughout. Difficult to follow.

Author Response

(The authors gave the same response as above.)

Round 2

Reviewer 1 Report

Comments and Suggestions for Authors

I would to thank the authors to take in account all my remarks and to have improve the manuscript in consequence. Now the manuscript is much clearer, better organized and the added information adds significant value at the paper. In this revised version the reader finds the information announced in the title and abstract. I congratulate the authors for this important work of revision and in particular for the MicroRNA and erythropoiesis part that is very interesting! Anyway, I detected still some imprecisions please review them.

“ T-cells, B-cells, and natural killer cells from lymphocytes and granulocytes, megakaryocytes” Pay attention lymphocytes is a general word to indicate Tcells, Bcells and NK cells they come from lymphoid progenitors

“The proerythroblast (ProE) is the earliest morphologically recognizable stage of an erythroid precursor and can be isolated from the whole MEP population” pay attention ProE is not a subpopulation of MEP is an independent differentiation stage cells

“supportive microenvironment, comprised of macrophages and stromal cells, within structures known as erythroblastic islands (EBI)” no paper talk about stromal cells in EBI, EBI are macrophages plus erythroblasts. 

“First observed by Marcel Bessis in 1958, the erythroblastic island serves as the primary location for erythropoiesis in mammals” little precision: terminal erythropoiesis

AML rare incidence? It is the most common kind of aggressive leukemia in adults

AEL paragraph! Pay attention following the new classification now there is not more M6a subtype that are classified in AML or MDS; AEL is only the old M6b subtype.

There is some therapeutic approach based of MicroRNAs involved in ineffective erythropoiesis?

Author Response

Thank you kindly for your review. 

Reviewer 4 Report

Comments and Suggestions for Authors

Much improved. Nice and comprehensive. 

Comments on the Quality of English Language

Much improved. 

Author Response

Thank you kindly for your review.